# A Revolution in Red Robes: Tibetan Nuns Obtaining the Doctoral Degree in Buddhist Studies (*Geshema*)

Nicola Schneider

Centre de recherche sur les civilisations de l'Asie orientale, 75005 Paris, France; schneidernicola@hotmail.com

**Abstract:** In the past, Tibetan nuns had no access to formal monastic education and thus could not obtain the two main diplomas and titles that are common in Tibetan Buddhism: the *khenpo* (*mkhan po*) degree in the more practice-oriented Nyingmapa school and the *geshe* (*dge bshes*) degree in the scholastic curriculum of the Gelukpa school; this essay traces the introduction of the Gelukpa study program in different nunneries based in India and Nepal in recent times; it addresses the question of gender asymmetry by showing the different hurdles that had to be overcome and the solutions, which have been found to allow nuns to become *geshema*s—the female form of *geshe*. Finally, I propose the first glimpse into the impact that the opening of higher Buddhist education to nuns has had and what this means for the future of the position of women in the religious sphere, as well as for Tibetan monasticism more generally.

**Keywords:** Tibetan Buddhist nuns; Tibetan female monasticism; Tibetan Buddhism; monastic education; *geshe* degree; gender asymmetry



"It is all mainly because of His Holiness the Dalai Lama's vision and his true care and kindness for his womenfolk".

(Rinchen Khandro Choegyal 2016)

## 1. Introduction

In the past, nuns from Tibet had only very few options to obtain training in Buddhist practice and knowledge. In most of the nunneries, they simply learnt how to read the scriptures, which they then had to memorise in order to participate in communal prayers and rituals. Few were also encouraged to go into retreat for meditation, which could last several years; however, higher Buddhist studies, such as philosophy and debate practice, were only taught in a handful of monastic universities belonging to the Gelukpa (dGe lugs pa) school, one of the four main schools of the Tibetan Buddhist tradition,[1] were reserved for monks. Equivalent institutions for nuns did not exist. And even if there were some exceptions, as for example, religious encampments—called *chögar* (*chos sgar*)—where nuns lived and studied alongside monks, these were mainly temporary in nature and mostly located in Eastern Tibet.[2]

In many Tibetan areas, becoming a nun by choice or sending a daughter to the convent was not something as systematic as it was for monks. Therefore, the number of nuns and their institutions in Tibet was—and still is—much smaller than that of monks. There were approximately 5400 monasteries with a little over 500,000 monks before the occupation of Tibet by China in 1951 compared to an estimated number of ca. 27,000 nuns and 700 convents.[3] In terms of proportion of the Tibetan population, it is estimated that between 10 and 12% of the male population became monks[4] and roughly 1% of the female population became nuns. Furthermore, there was an unknown number of nuns who stayed with their families—probably more than monks given the fact that in many regions there were no nunneries at all.[5] Even though the number of female monastics appears small at first look,

in comparison with other Buddhist countries, as well as compared with Catholic nuns in France, Tibet was home to one of the largest female monastic communities in the world.[6]

Since the religious revival in the mid-1980s, following the annexation of Tibet by China (in 1951), the flight of the Dalai Lama, the Tibetans' revered spiritual leader (in 1959), and the religious repression during the Cultural Revolution (1966–1976), education for nuns has markedly improved. Most nunneries in Tibet and in exile (mainly India and Nepal) have introduced study programs that go largely beyond reading and memorising, and several have adopted Buddhist curriculums that were traditionally taught in monastic universities. Not only that, some nuns have by now completed the entire Buddhist monastic course, thereby earning the highest diplomas and titles, which can be considered equivalent to 'doctoral' certifications in Buddhist philosophy or theology. There are now nuns who carry the title of *khenmo* (*mkhan mo*)[7]—the female form of *khenpo* meaning "professor", "scholar-abbot" or "preceptor"—in the Nyingmapa (rNying ma pa) school and the one of *geshema* (*dge bshes ma*)—the female form of *geshe* which translates literally "spiritual friend" (Skt. *kalyāṇamitra*) and refers to a "Doctor of traditional Buddhist philosophy." Whereas the former is bestowed after the completion of a course of approximately nine to twelve years, the latter needs a much longer period, twenty years and more; it is the Gelukpa degree and title that will be the main focus of this article.

The path to comparable education with monks was not without its pitfalls for nuns. Their generally accepted low status, the destructions of many if not most nunneries, and the financial burden of reconstruction and the founding of new institutions had to be shouldered before even thinking of how to introduce educational facilities.[8] Last, but not least, nuns faced challenges over hiring appropriate teachers, given the fact that there were none among their own communities and the general shortage of masters after a long period of religious repression. In addition, most monks would rather prefer to teach and stay in their monasteries, given their vow of celibacy.

The following article proposes to look at gender asymmetry in Tibetan monasticism with regard to education; it will trace the long journey that Tibetan nuns had to undertake before being able to take the examination for the highest degree of *geshema*. In doing so, it will shed light on the arduous negotiations that have taken place between the different stakeholders; moreover, it will draw on the sociological profiles of those nuns who have completed their studies, as they come from different countries and regions of the Himalayas that make up the Tibetan cultural area. The aim is to give a first glimpse of the impact that this opening up of Buddhist higher education for women may have on the future of Tibetan monasticism and, more generally, on the position of women in the religious sphere.[9]

## 2. Monastic Education in the Past

If most Tibetan monasteries offered at least a basic education to their novice monks, only a handful of them were specialised in providing higher philosophical studies and the prestigious degrees of either *khenpo* (Skt. *upādhyāya*) or *geshe* (Skt. *kalyāṇamitra*). As for the latter, from the Gelukpa school, these were the "three great seats" (*gdan sa gsum*) near Lhasa—the monasteries of Drepung ('Bras spungs), Sera (Se ra) and Ganden (dGa' ldan)—, as well as Tashilhunpo (bKra shis lhun po) in Shigatse and the monasteries of Kumbum (sKu 'bum) and Labrang (bLa brang) in northeastern Tibet, Amdo. Each of them housed several thousand monks, and were accordingly organised into semi-autonomous units called *dratsang* (*grwa tshang*) (Goldstein 2009, pp. 416–17). *Dratsang* can be translated as "college" as their structure resembled that of classic British universities like Oxford (Goldstein 1999, p. 20); hence also my term of "monastic university."[10] Drepung monastery, for example, had four of these "colleges," and Sera three. Not all the monks studying in one of these *dratsang*s were permanent members of the community; some joined from elsewhere, branch monasteries or different institutions, and some even from other Buddhist schools, in order to complete their higher education. Each college was independent from the main monastery, having its own financial resources, officials, teaching program, monks and its own abbot; however, there were times when the monks from different "philosophical colleges" (*mtshan*

*nyid grwa tshang*), and also from various monasteries, came together in order to train and even compete in debate (*rtsod pa*), the main component of the Gelukpa curriculum.

Tibetan monastic education has been compared to Western scholasticism as it is likewise a method of critical thought with its emphasis on interpreting great texts in a coherent system of logic and argumentation.[11] It has its origins in Indian Buddhist monasticism, and particularly the Nālandā tradition, and dates back to the later diffusion of Buddhism in Tibet (*phyi dar*), that is, after the tenth century when a renaissance started. At that time, new texts exposing the Mahāyāna tenets were brought from India, translated into Tibetan language and in many cases supplemented with commentaries by Tibetan masters. Over the following centuries, different schools were founded, and of these the Gelukpa was the latest to come to existence, in the early fifteenth century.[12] From the seventeenth century on, and with the aid of Mongol armies, it also became the dominant school in the country, with the Fifth Dalai Lama at its head, followed by successive incarnations.

The method of learning in the Gelukpa school consisted—and still consists—of three main components: memorisation, study, and debate. Students had first to memorise a given text, before getting the necessary explanation from their teacher; they then analysed and discussed the content during debate sessions, which usually took place in the afternoon and sometimes even until late at night. The duration of this curriculum was long, between twenty and twenty-five years and at the end, the candidate could take the examination in order to become a *geshe*, "Doctor of traditional Buddhist philosophy" or "Doctor of theology." However, not many Gelukpa monks completed their studies in the past, not only because it needed perseverance, but also because they had to sustain themselves during all these years, which was difficult for those who had no financial support from their families or sponsors.

As said before, there were no similar monastic universities for nuns in the past. And even in those few institutions where they had access to higher studies, as in the Dragkar Lama's religious encampments, debate practice was not part of their curriculum, contrary to the case of monks. However, we know also that in the mid-11th century, when there were six great monastic centers,[13] nuns and monks used to study together (Josayma 2017, p. 137). According to oral history, the latter were not as good in debating and often failed during competitions. It is said that Chapa Chökyi Senge (Phywa pa chos kyi seng+ge, 1109–1169), one of the earlier philosophers who had an influence on the development of logic in Tibet, thought that the whole situation of monks and nuns studying together was wrong and separated the two communities, thereby improving monastic rules, while cutting nuns off from the learning system (Ra se dkon mchog rgya 'tsho 2003, p. 82). Nevertheless, according to the Tibetan scholar Professor Thubten Phuntsok, from the Southwest University for Nationalities in Chengdu, nothing can be found in Chapa Chökyi Senge's collected writings or elsewhere to support this thesis. He thinks moreover that nuns were banned from debate later in history and that the prohibition was probably the work of the Kadampa or Gelukpa school.[14] Be that as it may, the result was that even when nuns had access to higher education, they were barred from an important part of the Gelukpa curriculum.[15]

A lot of changes have taken place in the last fifty years, not only for nuns, but also for monks, who face now fewer hurdles to finance their studies, at least in the Tibetan exile communities in India and Nepal. Monastic education such as we have seen has been developed in many other monasteries and even by Buddhist schools like the Nyingma-pas and Kagyupas that used to have their own scholastic traditions and were, generally speaking, more focused on the practice of meditation; moreover, since a decade or so, this type of traditional education has been introduced to some Tibetan schools for children. Monastic education has thus experienced a very strong institutionalisation in exile, among others thanks to the Dalai Lama, who regularly urges his compatriots in his speeches to deepen their knowledge of Buddhism through study, instead of simply reciting prayers or performing rituals. In particular with regards to nuns, he has actively pushed to improve

their educational level in contrast to their lower ordination status, which, as he says, cannot be solved by himself alone, but needs a larger consensus.[16]

I will now take a closer look at how nuns' education has come into focus since the late 1980s. In order to do so, I will draw on the example of Dolma Ling nunnery (sGrol ma gling), the first institute of higher Buddhist studies (*rigs lam slob gnyer khang*) for women; it is located near Dharamsala, a town in Northwestern India, which is also the seat of the Dalai Lama and of the Central Tibetan administration in exile. Most of the data come from my ethnographic work, which I started in 1996 and which I continued during my Ph.D. in the 2000s and later on in 2016–2017 when the first nuns passed their *geshe* degree examination.

### 3. Opening Up Education for Nuns

When Dolma Ling was founded by the Tibetan Nuns Project (*Bod kyi btsun ma'i las 'char*) in the beginning of the 1990s, only a handful of nunneries existed in the Tibetan exile community. Most of them were overcrowded and could not accommodate the many new nuns who had come from Tibet since the end of the 1980s; these were either fleeing the religious repression that had followed the demonstrations in Lhasa or had come from the far eastern part of Tibet in order to get an audience with His Holiness the Dalai Lama. Most expressed the wish to stay in exile and to study Buddhist philosophy, which they had not had the opportunity to do in Tibet. Thus, Ms Rinchen Khandro Choegyal, the then president of the Tibetan Women's Association, decided to found the Tibetan Nuns Project (TNP) as its branch in order to look after the nuns in particular.[17] With the help of the nun Lobsang Dechen and an American Buddhologist, Elizabeth Napper, support was secured from the Department of religion and culture, foreign sponsors, and the Office of His Holiness the Dalai Lama, the latter being especially encouraging. The aim was not only to improve the nuns' living conditions but also to develop monastic education for women. "I thought that the nuns need a purpose in life, and it is possible to give them one. Most of them want to study, to know more things, so the key is education," as Ms Rinchen Khandro expressed in an interview in 2001. At approximately the same time, Sakyadhītā ("Daughters of the Buddha"), an international association for Buddhist women, stated a similar goal during its first conference and founding occasion in Bodhgaya, in 1987; His Holiness the Dalai Lama, as well as the nun Lobsang Dechen having been present.[18]

A first group of about 100 nuns moved to Dolma Ling in 1994, which was still under construction then. During the next years, nuns would study in the morning and early afternoon and participate in the construction work afterwards (Figure 1). Most of them were from Tibet, but some also came from the Indian Himalayas, mainly Ladakh and Spiti, respectively regions with only a few and no nunneries at this time. In the following years, their number expanded dramatically, to a little more than 200 at the beginning of 2000; it has been stable since then, with a proportion of approximately 75% of nuns from Tibet and 25% from India, Nepal, and Bhutan. Some nuns and laywomen from Western and other Asian countries have also joined Dolma Ling to study but usually they stay only for a short time. Since the Tibetan uprising in 2008 and the subsequent closure of the borders between China and India, only very few Tibetans have been able to come to exile and this is also the case for nuns, leaving more places in the convent to those from Himalayan areas. Meanwhile, many have also left the institution since completing their studies, making it possible to recruit around twenty new nuns every year.

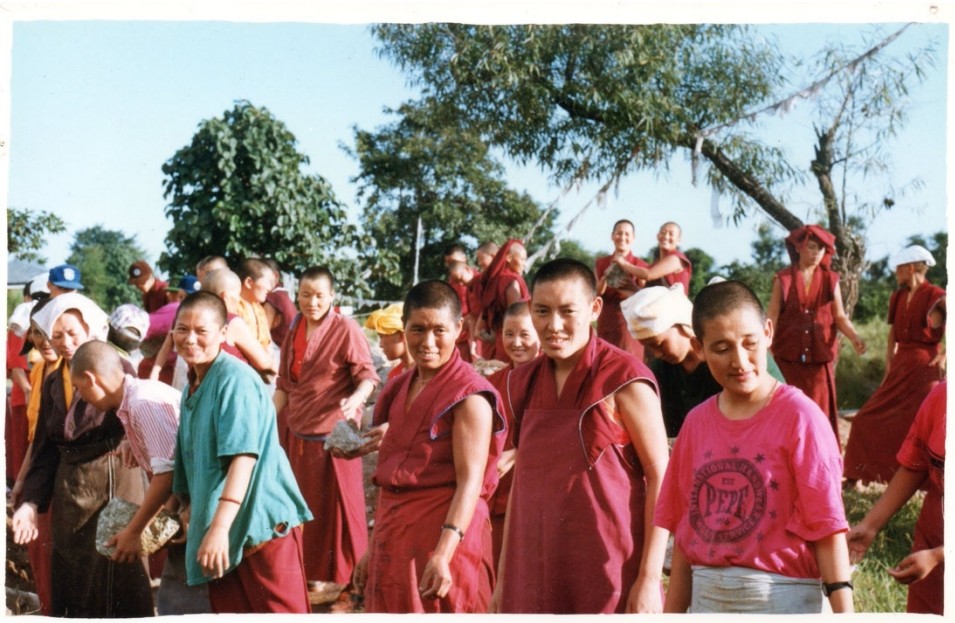

**Figure 1.** Nuns from Dolma Ling participated in the construction of their nunnery (1996).

In the beginning, Dolma Ling was envisioned as an eclectic or non-sectarian nunnery and Institute of Buddhist studies.[19] However, the curriculum that was finally implemented is clearly the one from the Gelukpa school, although supplemented by teachings from the three other schools; it follows the model of the Institute for Buddhist Dialectics (*mTshan nyid grwa tshang*), a new monastic centre of learning, which has implemented the traditional education of the "three great seats" while also being innovative in training their monks in order to allow them to adapt to life in modern society.[20]

Because in the early days, most nuns in Dolma Ling were not literate when they arrived, the institute set itself the goal of providing them not only with traditional monastic education, but also with basic secular education.[21] Thus, the curriculum is divided into two parts: (1) secular subjects such as Tibetan language, Tibetan history, social sciences, mathematics and, recently, physical sciences[22] and (2) monastic education. Most courses take place in classrooms, much as in a school, except for the practice of debate, which takes place in the open air. I will come back to this later. After about ten years, the nuns receive a first diploma, called Prajñāpāramitā Buddhist Philosophy (*phar phyin*), which is equivalent to a BA and allows the students who so desire to continue their studies in other Tibetan institutions of higher education order to become, for example, teachers of Tibetan language, English or social sciences. Many nuns excel in the Tibetan language and have found jobs in schools and nunneries alike. Others have decided to work in administration or communication.[23]

At the heart of monastic education is the curriculum of the Gelukpa Buddhist school that leads to the degree and title of *geshe* or "Doctor of traditional Buddhist philosophy." It is based on a set of texts—from the so-called "Five great texts" (*gzhung chen bka' pod lnga*)—gathered in a textbook or written charter (*yig cha*) and organised in a number of topics; these include metaphysics, logic, monastic rules, and, most importantly, philosophy according to the *Mādhyamika* or "Middle Path" system, which goes back to the Indian philosopher Nāgārjuna; it is a system of epistemological argumentation. As we have seen, the Gelukpa curriculum has three complementary components—memorisation, study, and debate. In Dolma Ling, it is practised as follows. In the morning, the nuns have Buddhist philosophy classes (among others) during which they read a given text with the teacher who then explains the meaning of the content orally. Most religious texts are written in the so-called "religious language" (*chos skad*) that is difficult to understand even for literate

Tibetans without appropriate formal training. The aim is to train students in Buddhist logic and analytical reasoning by showing them specific examples during these lessons. The same text will then be learnt by heart before being discussed again in class. The memorisation of the key texts is an important part of the study and students must learn thousands of pages before they can excel in debate.[24]

Then, in the afternoon, the actual debate or dialectical session takes place in the debate court (*chos ra*); it relates to the subjects and questions already discussed in class. After a short prayer to Mañjuśrī, the bodhisattva associated with wisdom, students form groups of two or three people, and even up to ten or more for beginners; they then distribute the roles: the *defender*(s) (*dam bca' ba*), facing one or more standing *questioner*s (*rigs lam pa*); it is up to the defender, who then takes her seat, to launch the debate by presenting a thesis, which, by definition, must be accurate, that is, corresponds to the Buddhist world view. Her role is to defend it, whatever the cost; this is followed by a preparatory phase, during which the two parties determine the starting point and the terms of the debate. Once the agreement has been reached, the main part of the debate can begin; it is now up to the questioner(s) to find a way to refute her opponent's thesis. To do so, she proceeds by enquiring, formulating her questions in such a way as to force the defender to contradict herself. The latter, for her part, must try to thwart any attempt at attack by choosing the answer that she can defend and that does not contradict the basic thesis, which would lead to defeat.[25]

Every debate is also accompanied by gestures that give it a performative, even theatrical dimension (Figure 2). The standing questioner plays the main role: after each statement, she claps her hands and simultaneously stamps the ground with her left foot. Through body movements and the raising of voice, she not only mobilises her own intellectual abilities, but also tries to destabilise her opponent.

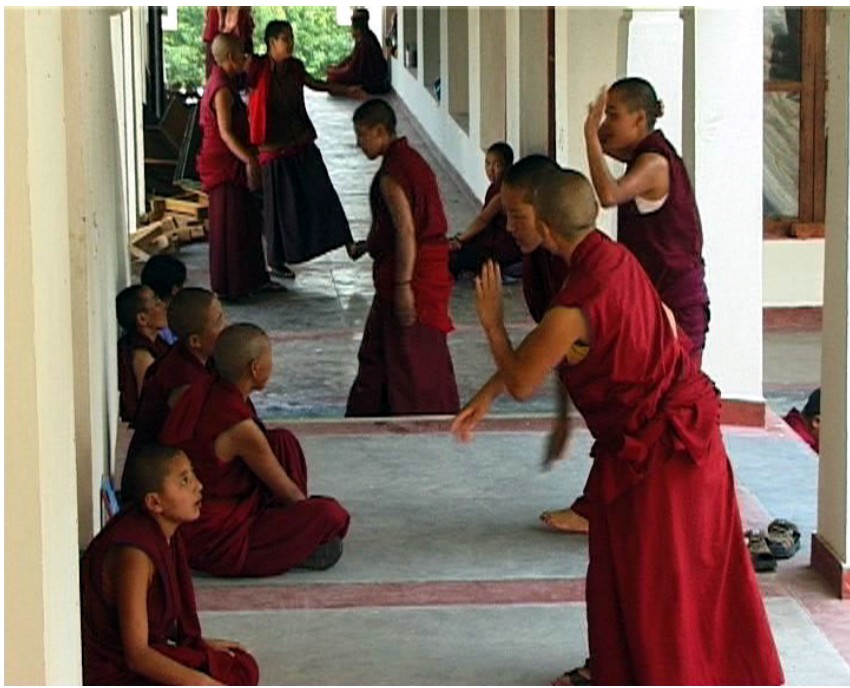

**Figure 2.** Debate training in Dolma Ling.

The practice of debate constitutes the essence of Gelukpa monastic education. In Dolma Ling, the nuns debate for approximately one and a half hours per day, and a little more for the most motivated; this is less than in the male monasteries: in Sera, for example, monks debate up to four hours per day; however, there are also periods when more intense debate sessions take place in Dolma Ling: once a month, an inter-class competition is organised, for which the students prepare for a whole week throughout the day with no

other classes or rituals being held in the meantime. In addition, there is a competition between different nunneries once a year; this kind of event, called "winter debate [of the village Jang]" (*'jang dgun chos*), is a traditional custom in which many monasteries around Lhasa took part.[26] It was introduced into exile nunneries in 1995. Since then, each year, nuns chosen from among ten different institutions located in India and Nepal participate in the event.

Although education at Dolma Ling is diverse and nuns can specialise in many different fields—such as administration, communication or handicrafts—, the practice of debate is highly valued. The students' results are mainly judged by their ability to debate. All those of the same nunnery know the best debaters among them, even though there are generally only very few.

## 4. Seeking Recognition

The Gelukpa curriculum was introduced to Dolma Ling and other nunneries in order to allow nuns to have access to the same scholarship as monks. In order to succeed, it was also necessary that they get the possibility to pass the same diploma leading to the title of *geshe* or "Doctor in traditional Buddhist philosophy." For a long time, those responsible for the Tibetan Nuns Project involved in the process were unsure how to realise this goal. The main concern was one particular part of the curriculum: the *Vinaya* or monastic discipline (see Table 1). Studying the *Prātimokṣa* (Tib. *so so thar pa*) part of the *Vinaya*—the text containing the rules for monastics—requires theoretically that the monk, or in our case the nun, has taken all the precepts, that is, he or she is a *gelong* or a *gelongma* (*dge slong* [*ma*])—the Tibetan equivalent of a *bhikṣu* or a *bhikṣuṇī*. Because full ordination was not conferred on women in Tibet, it was thought nuns could study the *Vinaya* only partly and not in its entirety since ordination functions here as a preliminary and strictly necessary preparation or initiatory process before approaching the *Prātimokṣa*.[27] Not being able to study the monastic discipline as required would mean that nuns would not have completed their curriculum, in contrast to monks; it was a vicious cycle from which there seemed to be no way out.[28] The result was that many nuns, not being able to pursue their studies, began to reorient themselves by going into retreat, starting to teach in schools, or leaving the monastic life altogether.

**Table 1.** The Gelukpa monastic curriculum in Dolma Ling.

| Class | Number of Years |
|---|---|
| Preliminary studies (including *Pramāṇa*; Tib. *tshad ma*) | 4 |
| "Perfection of Wisdom" (Skt. *Prajñāpāramitā*; Tib. *phar phyin*) | 7 |
| "Middle Path" (Skt. *Mādhyamika*; Tib. *dbu ma*) | 3 |
| "Phenomenology" or "meta-doctrine" (Skt. *Abhidharma*; Tib. *mngon pa*) | 3 |
| **"Monastic discipline" (Skt. *Vinaya*; Tib. *'dul ba*)** | 1 |

In 2011, unexpected news came: it was announced that for the first time in history, a nun of the Tibetan tradition would be honoured with the diploma and title of *geshema*; however, Kelsang Wangmo, as she goes by her ordination name, is a German national and had studied at the Institute for Buddhist Dialectics that usually welcomes only monks.[29] As she was not fully ordained either, she was advised to study other texts than the *Prātimokṣa sūtra*, while also being allowed by her teacher to listen to the teachings given to her monk co-students recorded for this purpose. It was a surprise, not only for the Tibetan Nuns Project who had faced this issue for some time, but also for the Tibetan population. The very idea of nuns one day becoming *geshema*s was already accepted at this point with the feminisation of the initial masculine title being used widely. But the fact that the first *geshema* was a foreigner also elicited some criticism, by Tibetan nuns as well as by others.[30]

Following the announcement, the Tibetan Nuns Project started a new campaign directed at some of the great religious masters and the Department for religion and

culture—the official body that manages monasteries and nunneries but not necessarily their study programs—in order to find a solution for the many Tibetan nuns who were waiting to proceed with their studies; it was decided to include the topic to the agenda of the eleventh Tibetan religious conference, which took place in September 2011 (Central Tibetan Administration 2011; Phayul 2011). Whilst only the heads and representatives of the four major schools of Tibetan Buddhism and the Bon tradition, which are all male, assemble at these meetings, the Tibetan Nuns Project prepared a written statement in advance explaining its support for nuns to obtain the *geshema* degree.[31] The declaration was read aloud and provoked a discussion on the knowledge level of nuns. Several speakers present pronounced themselves vehemently against a decision to confer the *geshema* title, arguing that nuns only study three out of five courses required by the curriculum. By saying this, they meant that nuns have not studied the entire *Vinaya*, which was indeed true, but also that they have not completed the *Abhidharma* class, which was a major misunderstanding. When Rinchen Khandro Choegyal, director of the TNP, heard about what had happened during the eleventh Tibetan Religious meeting, she got upset. How was such a misunderstanding possible after all these years and discussions with so many religious dignitaries? She first required an answer from some of the *geshe*s with whom she was in contact and who were present at the meeting. She then went directly to Pema Chhinjor, the then Minister of the Department of religion and culture, and a former subordinate when she was the education minister (1995–2001), but also a longtime friend of her family.[32] The latter apologised saying that he did not know exactly about the nuns' learning; they discussed how to bring up the issue again and agreed to organise a further meeting the following year with the heads of the three monastic universities in South India (that is Sera, Ganden, and Drepung), a representative of the Department of religion and culture, and one from the Tibetan Nuns Project. Meanwhile, different nunneries' representatives also assembled to discuss further steps to take.

During the Tibetan New Year festivities in February 2012, some Tibetan nuns were invited to debate in front of the Dalai Lama at the main temple; it is said that His Holiness was delighted to see the nuns' progress in debating and personally asked the Department of religion and culture to push the issue of *geshema* (The Tibet Express 2012; Phayul 2012). One month later, on 8 March 2012, during a meeting with the Department of religion and culture, the Dalai Lama asked to formulate a proposal, which specifies the requirements and modes of examination that nuns should undergo to obtain the title of *geshema*. The Department contacted the Gelukpa council—responsible for all affairs concerning the Gelukpa school—and the *Gandentripa*—head of the Gelukpa school—in order to proceed; however, the latter was not available since he was abroad at that time. He proposed instead to meet and talk about the issue at the beginning of the following month, when he would himself be coming to Dharamsala. Meanwhile, Samdhong Rinpoche, former Chief Minister and a *geshe* himself, also gave his agreement to go ahead and finalise the constitution. Both the *Gandentripa* as well as Samdhong Rinpoche are considered by the Tibetan Nuns Project and the Department of religion and culture to have been very supportive of the issue.

Finally, the Department of religion and culture called a big meeting on 18th and 19th of May 2012 with the objective of reviewing the draft proposal, refining and finally approving it.[33] In attendance and as signatories to the constitution there were representatives of the Tibetan Nuns Project, of the Institute of Buddhist Dialectics, of the College for Higher Tibetan studies, as well as two senior nuns and a *geshe* professor from each nunnery where the Gelukpa curriculum was introduced (Jangchub Chöling, Khachö Gakyil Ling, Geden Chöling, Jamyang Chöling[34] and, of course, from Dolma Ling). The result is a charter or constitution (*sgrig gzhi*) called "Rules concerning the examinations and acquisition of the title *geshema* for nuns who have accomplished the study of the five great texts in the Tibetan nunneries and academic institutions."[35] The content is presented in eight chapters, five pages long, plus a blank "Certificate of *Geshema* degree" to be signed by the Department of religion and culture and examiners. The constitution stipulates that the length of study has to be at least seventeen years; the minimum percentage to be obtained each year is

75%; and the period of revision and examinations has to last four years. The content of the curriculum and the tests is also fixed,[36] as is the composition of the group of supervisors who preside over the conduct of the examination.[37]

What about the *Vinaya* part of the curriculum, the subject which had been the focus of so many polemics up to then? Geshe Rinchen Ngödrub (*dge bshes* Rin chen dngos grub), a scholar specialising in monastic discipline, who had been teaching in Dolma Ling for several years, proposed to design a new program for this part of the studies. Being one of the defenders of full ordination for nuns and the author of a book based on his many research findings on the subject (See (Ser byes lha rams ngag gi dbang phyug Rin chen dngos grub 2007)), he decided to build up an agenda founded on Indian classical root texts and an auto-commentary (*rang 'grel*) by Śākyaprabha.[38] Unlike so many other texts, the latter had not been further commented by Tibetans until he himself had recently drawn up a commentary in thirty paragraphs. He had already used it as a teaching tool in Dolma Ling and it had been also distributed to all the other nunneries in exile. His proposal was astute: knowing that the Tibetan monastic curriculum draws mostly on commentaries by Tibetan scholars, his suggestion of returning to the original, supposedly more authentic text would probably confer more prestige on the nuns, in the absence of full ordination. He was, of course not the main decision maker, but had consulted several other *Vinaya* scholars, among them Geshe Kesang Damdül (*dge bshes* sKal bzang dgra 'dul) and Jamphel Dragpa ('Jam dpal grags pa) from the Institute of Buddhist Dialectics; both appreciated the importance given to Indian scholasticism and confessed to him that they would like to see a similar program in monasteries.[39]

After the Dalai Lama's intercession, an official approval from the big monastic universities in the South of India was deemed to be unnecessary. Likewise, the *Gandentripa* finally agreed by phone; it is interesting to note here that both the Minister of the Department of religion and culture, as well as the head of the Gelukpa school,[40] had been appointed recently, in 2011 and 2009 respectively. Thus, the *geshema* issue was resolved thanks to the Dalai Lama, but also partly after a series of political changes and because of a new generation of monk scholars like Geshe Rinchen Ngödrub.

### 5. Going for Examination

The meeting was a success, and many nuns who had previously "disappeared" returned to their respective nunneries to participate in the project. In 2013, the first round of examinations took place, with the participation of twenty-seven nuns from five different nunneries.[41] In 2016, a total of twenty nuns, including six from Dolma Ling, passed their final exam; some had failed intermediate steps or dropped out altogether.

The final exam was organised in one of the participating nunneries, Geden Chöling, at the beginning of May 2016; it consisted of debate sessions (Figure 3), written examinations—in Buddhist philosophy, but also in Tibetan grammar, history and in Western science—, and the oral defense of an approximately fifty-page-long dissertation on subjects dealt with in the Five Great Treatises—the main corpus of texts studied by Tibetan monastic students—which had to been handed in beforehand. To make these examinations credible in the eyes of the Tibetan population and in particular of the clergy, its organisation was entrusted to the Department of religion and culture, whereas the topics were developed by *geshe*s from the three monastic universities located in southern India. Likewise, the invited auditors came from the "three great seats." Most were quite young graduates themselves and several of them were from the same region in Tibet, Kham, as some of the nuns taking the examination.[42]

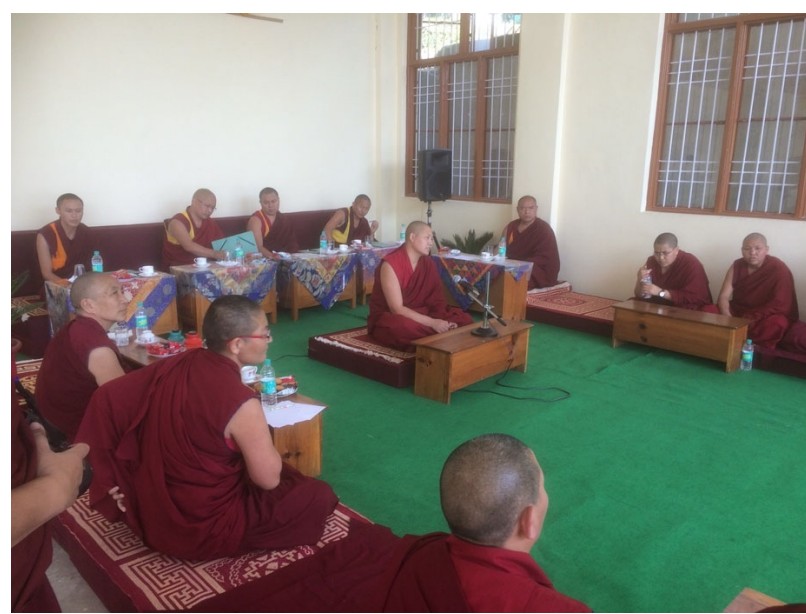

**Figure 3.** Debate test during the final exam for the *geshema* degree.

As for the sociological profile of the nuns who passed the final exam: eleven were from Tibet; six came from Tibetan-speaking Indian regions where Tibetan Buddhism is traditionally practiced (one from Spiti, one from Ladakh, two from Zanskar, and two from Kinnaur), two from the Tibetan-speaking enclave of Mustang (Nepal) and one from Bhutan. All come from families with farming or pastoral background, that is, populations who live in the Himalayan countryside where access to education is still very rudimentary, especially for girls. All nuns had also faced many obstacles before being able to study Buddhist philosophy: many had no nunneries in their native regions, which has led them to move—for some of them far away from their original home; a few Tibetans had been banned from staying in a nunnery since the quotas for admission in contemporary Tibet are very restrictive; and most have faced a lack of economic support. Nuns from Dolma Ling were not the only ones who had spent a considerable amount of time building their nunnery. In addition, all have had to carry heavy administrative burdens in their respective institutions, from which monk students are generally exempt. Therefore, most nuns had started their studies at a relatively late age and made frequent temporary interruptions, which explains why this first group of graduate nuns had an average age of about 43 years—the youngest being 36 years and the oldest 50.

The results of the first *geshema* exam were announced in July 2016: all the twenty nuns who participated passed and three of them did so with distinction. The diplomas were awarded in December the same year at an official ceremony that was organised in Drepung monastery in parallel with the celebrations of the 600th anniversary of its founding. The Dalai Lama personally handed the nuns their certificates in the presence of many religious dignitaries, monks and nuns from the various monasteries in the area, as well as representatives of the Tibetan administration in exile (Figure 4). Since the monastic university of Drepung is located in Mundgod, one of the major Tibetan settlements in exile, the organisers of the ceremony have assured a maximum of Tibetans attending the event and thus recognising the nuns' achievements.[43]

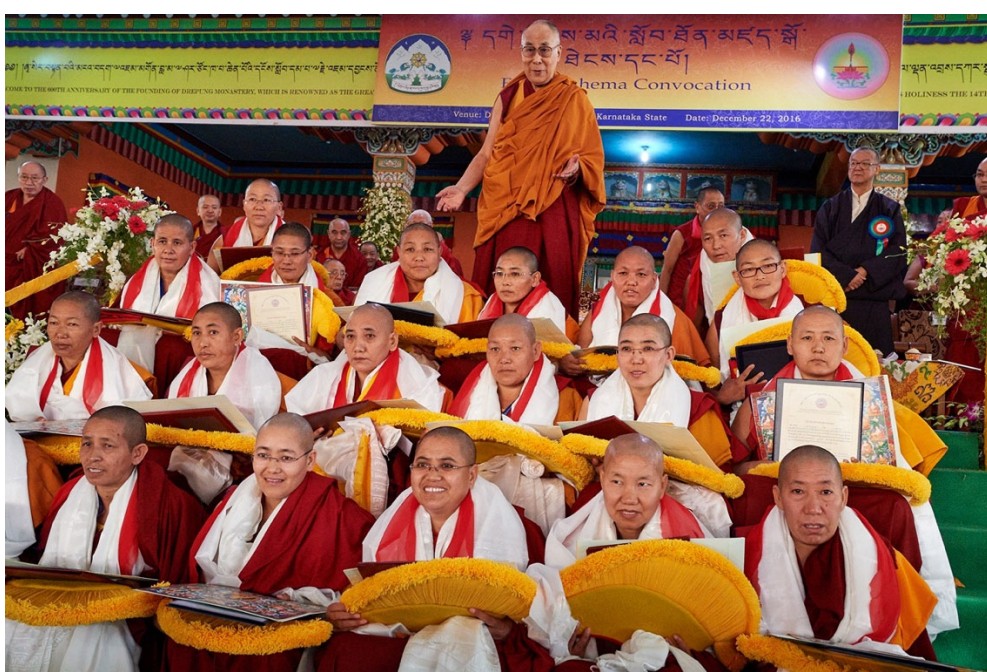

**Figure 4.** The graduation ceremony of the first group of *geshema*s (Photo courtesy Olivier Adam).

However, although nuns had now followed a similar curriculum to the monks and a suitable solution was found for the issue of the study of monastic discipline, rumours spread about their *geshe* status being considered as "low" (*chung ba*). In the Tibetan monastic hierarchy, there exists a distinction between four different *geshe* degrees, the highest possible one being the *geshe lharampa* (*dge bshes lha ram pa*); it is awarded by the three great monasteries only to a minority of selected monks after a revision and examination period of six years (instead of four). Furthermore, after their graduation, many monks take up tantric courses in one of the two specialised monastic colleges Gyuto (rGyud stod) and Gyumé (rGyud smad), also reserved for men. In order to allow the nuns to meet as many requirements as possible, the Tibetan Nuns Project, supported by the Dalai lama, negotiated with the former, situated close to Dolma Ling in the valley of Dharamsala, to accept nuns as well; it was finally decided that the most convenient way was to regroup all the *geshema*s in Dolma Ling and to invite teachers from Gyuto in order to instruct them. Thus, a first course of tantric studies was initiated in late 2017 for a duration of one year. Twenty-three *geshema*s took part and received a certificate from the Minister of the Department of religion and culture, Venerable Karma Gelek Yuthok (Karma dge legs g.yu thog), at the beginning of 2019 (Central Tibetan Administration 2019).

Between the years 2017–2019, twenty-three more nuns passed their *geshema* exams with success: six in 2017, ten in 2018, followed by seven in 2019. Because of the Covid pandemic, examinations had to be postponed during the last two years (2021–2022), but have resumed in August 2022.[44]

Dolma Ling has produced this far sixteen *geshema*s. Two of them who have obtained top scores, Tenzin Künsel and Delek Wangmo, were hired as teachers in 2019; they are teaching junior Buddhist philosophy classes and supervise the debate sessions; moreover, they have decided to take responsibility for the many new young nuns from Ladakh, Zanskar, and Nepal, who joined Dolma Ling in the winter of 2022 because of their respective nunneries being emptied after the pandemic; they are preparing them especially in the Tibetan language in order for them to be able to follow the Buddhist philosophy courses. Other *geshema*s have chosen to go into retreat, to work on research projects launched by the Tibetan medical institute and the Central Tibetan Administration, to deepen their knowledge of Western science by taking up courses in the exchange program with the University of Emory, to lead meditation courses for laywomen, or to give tuition for

children during school holidays; it turned out that the new degree has opened up many new possibilities for nuns.

## 6. Female Emancipation and Monastic Education

Even though it might be a little early to draw conclusions on the nuns' change of status after becoming *geshema*s, a few remarks can already be made.

A first point is the reception of these nuns by their families and the society. As soon as the information about the examination results came out, several families of the nuns, especially those from India, set out in order to celebrate the event, even before the official graduation ceremony had taken place; they thus expressed the pride with which they received the news. Families from Tibet were also keen to let their nun members know about the joy they felt for them by congratulating them through social media or sending relatives from India to take part in the celebrations in the respective nunneries, which lasted over several months.[45]

When visiting some of the nunneries involved in the Buddhist philosophy education program during the following years, I have also noticed that these *geshema*s have acquired a new status among their sisters: previously considered simply as elders, they now receive significant respect in their respective institutions. Young nuns treat them with the same reverence as male religious masters, that is, they serve them and bow in front of them—the Tibetan way of showing respect. In the temple of Dolma Ling, *geshema*s now have a special seating arrangement, with higher places than those reserved for the other nuns—even though male teachers are always seated slightly above them; they no longer must participate in menial chores such as cooking or cleaning. By contrast, two of them were elected to the office of disciplinarian (*dge bskos*) and general manager (*phyag mdzod*), the two highest positions generally attributed to nuns in their institutions; the abbot, or in the case of Dolma Ling, the principal, is and has always been a monk or a layman.

Generally speaking, the *geshema*s are also highly valued as teachers by other nunneries and by Tibetan schools, where some have already started to work in recent years; however, for the time being, the watchword is that they have first to serve their original nunneries, which are also very much in need of female teachers.[46] Most importantly, *geshema*s have become role models for young nuns. And they take this very seriously, for example, by regularly addressing the younger generation in order to encourage them in their studies; this is also significant for them, since, as some have pointed out to me, they did not have any examples to follow when they embarked on this type of study.

In recent years, as mentioned above, the curriculum of Gelukpa monastic education has been introduced into many monasteries and nunneries in exile. The same is true for Tibet. In several Gelukpa nunneries in Kham, we can now find nuns who are preparing the *geshema* degree while learning debate.[47] More surprisingly, perhaps, nuns of the Bon tradition, which is distinct from Buddhism but shares many characteristics when it comes to monasticism, were also awarded the *geshema* degree.[48] In October 2018, a ceremony took place in honour of nine Bonpo nuns from the three different regions of Tibet (Ü-Tsang, Kham, and Amdo) who had passed their *geshema* degree at Khyungmo (Khyung mo) nunnery in Thika (Khri ka), Amdo ('Chi med g.yung drung 2018). The initiator and main teacher was Geshe Söpa Gyurme (*dge bshes* bSod pa 'gyur med), who studied at the famous Bonpo monastery of Menri (sMan ri) in India.[49] It is quite possible that he had established a similar curriculum of either nine or thirteen years of study in Bonpo monasteries and nunneries in Tibet.[50] More recently, the only Bonpo nunnery in exile, Ratna Menling (Ratna sman gling), bestowed the *geshema* degree for the first time on five nuns, four from Tibet and one from Dolpo (Nepal) (Central Tibetan Administration 2022). As for those from India, debate was part of their academic program, which covers a total period of eleven years.[51] The *Vinaya* part of the study, which also takes a year, was not contentious, because Bonpo nuns can choose to become fully ordained: they are called *drangsongma* (*drang srong ma*) and follow a total of 360 precepts (Roesler 2015, p. 436).

I already mentioned the *khenmo* degree, which has been awarded to nuns by the Nyingmapa school in Tibet. Contrary to the Gelukpa school, the latter had taken the decision to open education and diplomas to female monastics very early on, at the beginning of the 1990s. The initiative had begun in the religious encampment of Serthar Larung Gar, founded by one of the greatest contemporary lamas and religious revivalists, Khenpo Jigme Phüntsok (mkhan po 'Jigs med phun tshogs, 1933–2004). His niece, Mumé Yeshe Tsomo (Mu med ye shes mtsho mo, b. 1966) or Mumtsho, as she is affectionally called by her disciples, was among the first group of nuns to receive the *khenmo* degree. Being the abbess of the Pema Khandro Duling (Padma mkha' 'gro'i 'du gling) nunnery, located inside the religious encampment, and recognised as a living *khandroma* (mka' 'gro ma), a female incarnation and saint, she can be considered as one of the greatest Tibetan Buddhist nuns of our time (Schneider 2013, pp. 156–61; 2015). Over the years, the academic program for nuns in Larung Gar has been refined and institutionalised, now taking up to fifteen years for "cultural studies" (*rig gnas*; here mainly Tibetan, English and Chinese language) and Buddhist philosophy studies and up to thirty years for those who continue with the tantra section (*rgyud sde*) and oral instructions into the lineage of Longchenpa (kLong chen pa, 1308–1364) (Padma'tsho/Baimacuo, pp. 11–12). In 2021, more than a hundred nuns held the *khenmo* title, a milestone in terms of female monastic education (Ibidem, p. 9). Many of them are now teaching younger nuns in Larung Gar, but also in various other nunneries of eastern Tibet; moreover, nuns have started to participate in research and editing projects, the publication of the *Ḍākinīs Great Dharma Treasury* (*mKha' 'gro'i chos mdzod chen mo*), a collection of fifty-three volumes on Buddhist women from Mahāprajāpatī to Mumtsho, being a major achievement and new endeavour of female scholarship.[52]

A similar development can be observed in at least one other Nyingmapa nunnery in eastern Tibet, Tashi Gönsar (bKra shis dgon gsar), albeit in a more modest way.[53] Since 2011, seven nuns have received the *khenmo* degree; their study program was more informal, based mainly on teachings and initiations given by their lama and different invited religious masters over many years during the bi-annual religious assembly, and a complete course in Tibetan medicine, organised on the premises of the convent; these *khenmo*s are now instructing in a more organised way, i.e., in classes designed for younger nuns. Some are also involved in the teachings during the religious assemblies that several hundred nuns, monks and lay people attend in order to deepen their Buddhist knowledge. Furthermore, several of them are working as doctors in the Tibetan medical clinic, which is run by the nunnery.

Meanwhile, Nyingmapa nuns living in India and Nepal are still waiting to get the official approval to become *khenmo*s. For several years now, many nuns from Tsogyal Shedrupling nunnery (mTso rgyal bshad sgrub gling) in South India and Shugseb (Shug gseb) nunnery in Dharamsala have finished their nine-year course, usually required for monks before taking the final examination; however, nuns have been only given the diploma and title of *lobpön* (*slob dpon*) or "teacher," an inferior qualification and designation. Here too, it seems that because of their lack of full ordination, Nyingmapa dignitaries have so far been hesitant to bestow the degree,[54] but it looks as if there will be an alternative solution in a very short time.[55]

Last but not least, the Sakyapa school in India recently announced the graduation of three nuns as *khenmo*s.[56] The degrees were awarded after ten years of rigorous studies containing in particular the "Eighteen renowned scriptures [of Sakya philosophy]" (*grags chen bco brgyad*) and four years of teaching experience.[57] The decision of bestowing the *khenmo* degree to nuns was taken by a committee of Sakya scholars who consulted with Sakya Tridzin (Sa skya khri 'dzin, b. 1945), 41st throne holder of the Sakyapa school.

## 7. Conclusions

For more than thirty years now, Tibetan nuns in exile have started to engage in higher Buddhist studies previously reserved for monks. The journey to become a *geshema* has been long and full of obstacles; however, there were also many people who helped the nuns,

first and foremost the Dalai Lama and the managers of the Tibetan Nuns Project, but also, of course, many monks who were mobilised as monastic allies. When talking about the new *geshema*s, the Dalai Lama modestly says "that is my small contribution,"[58] thereby indicating how important the establishment of proper education for nuns has also been for him. His continued commitment to nuns' monastic education clearly stands in contrast with his timid support for their full ordination.

The opening up of religious education for nuns has significantly changed the status of women in Tibetan monasticism and Tibetan societies, in Tibet itself, in exile, as well as in the Tibetan-speaking parts of the Himalayas. In the near future, nuns will no longer depend on monks for teaching and administrating their institutions. In the eyes of laypeople, they now deserve more respect, which, in turn, has translated into more social support and esteem than they used to get in the past; moreover, nuns are now able to contribute better to society by teaching, instructing, and also counselling lay people, especially lay women.

In the absence of full ordination, one cannot say that there is parity or equality between monks and nuns, but a great step towards empowerment has been taken. By persevering seriously in their studies, nuns have shown their capacity to engage in the same type of higher Buddhist education as their male counterparts; it will now be up to them to continue and preserve the tradition of Tibetan scholasticism.

The question also remains how important the ordination status actually is in the modern context where degrees are more and more institutionalised. As one of the contemporary scholars of Tibetan religion and culture, Khenpo Tenkyong, reminds us: the most famous *geshe* in Tibetan history was Dromtönpa (whose full name is sBrom ston rgyal ba'i 'byung gnas, 1004–1064)—chief Tibetan disciple of Atiśa (982–1055?) and founder of the Reting (Rwa sgreng) monastery (Gardner 2010)—who was not a monk but a lay devotee (*dge bsnyen pa*; Skt. *upāsaka*), renowned for his teachings on monastic precepts.

Some further challenges always persist, like the fact that nuns are seen as "small" *geshe*s compared to seemingly "full-fledged" monk *geshe*s. Some people continue to propagate malicious gossip, suggesting that nuns might not be as good in debating as monks or that they have not thoroughly understood the content of Buddhist philosophy. Another hurdle is their participation in politics, which, in a cultural system where politics and religion are closely intertwined, was and is always in some regard the prerogative of monks. During the recent 2021 elections in exile, one of the *geshema*s from Dolma Ling, Delek Wangmo, was appointed as election commissioner, the first time that a nun has held such a position (Tenzin Dharpo 2020). However, it seems to me unlikely that a nun will be elected as a parliamentarian in the near future, even though monastics, male and female, have two voices and ten reserved seats to be distributed among the four Buddhist schools and the Bonpos.

Thus, with regard to monastic education, the traditional gender asymmetry is always prevalent, but by tackling unequal access to religious instruction, it is also slowly being dissolved. The success of the first *geshema*s has clearly inspired the other Tibetan Buddhist schools, who, in turn, have been quick to act by also awarding degrees to their nuns.

**Funding:** The research for this article has been made possible thanks to funding received from the Centre de recherche sur les civilisations de l'Asie (CRCAO), the École des hautes études en sciences sociales (EHESS) and the École française d'Extrême-Orient (EFEO).

**Acknowledgments:** I am grateful to Charles Ramble, Nicolas Sihlé, Ester Bianchi, as well as for the anonymous reviewers for their willingness to read an earlier draft of this article and for their suggestions and corrections.

**Conflicts of Interest:** The author declares no conflict of interest.

## Notes

1　The other three are: the Nyingmapa (rNying ma pa), the Kagyüpa (bKa' rgyud pa), and the Sakyapa (Sa skya pa) schools.

2　In contrast to monasteries (*dgon pa*), most religious encampments only existed for a short time and were then either dissolved or transformed into proper monasteries. Examples of religious encampments which proposed higher Buddhist studies for monks

and nuns were those founded by the Third Dragkar Lama from the Gelukpa school (Schneider 2011) and Adzom Gar (A 'dzom sgar), from the Nyingmapa school, situated in the Tromtar (Trom tar) region of Kham (personal communication from the nun Sherab Wangmo, born 1947). Contemporary religious encampments such as Larung Gar (bLa rung sgar) and Yachen Gar (Ya chen sgar) tend to exist for longer time.

3   These statistics were issued by the Central Tibetan Administration in exile few years after the flight of the Dalai Lama followed by many Tibetans to South Asia. There exist different estimations of the number of monks and nuns and their institutions (e.g., Goldstein 2009; Jansen 2018; Ryavec and Bowman 2021), but they only refer to Central Tibet, leaving out Tibetan populations living in the two eastern regions of Amdo and Kham.

4   It must be emphasised that these numbers only represent a very rough estimate. See Goldstein (2009, p. 411); Samuel (1993, pp. 578–82) precises that these number only concern centralised Tibetan areas.

5   My estimate if we take the 27,000 nuns stated in the statistic of the Central Tibetan Administration for granted; and even more if we add an approximate number of "household" nuns. Ryavec and Bowman (2021, p. 209) suggest a proportion of 6% of the female population, but as already stated, this cannot apply to the whole area populated by Tibetans, especially because nunneries were only very rare in Kham and Amdo (Schneider 2013).

6   For Thailand, another important Buddhist country, Stanley Tambiah (1976, pp. 266–67) estimates the number of monks as 1–2% of the male population. As for Catholic nuns, Langlois (1984, p. 39) estimates that nearly 1% of the female population were nuns—mostly congregationalists—at the peak in 1880.

7   For more information on the *khenmo* degree as bestowed mainly in the religious encampment Serthar Larung Gar (gSer thar bLa rung sgar) in Tibet, see Schneider (2013, pp. 153–61), Liang and Taylor (2020) and Padma'tsho/Baimacuo (2021).

8   Concerning the financing of convents in another context, Zanskar, see Gutschow (2004, pp. 77–122).

9   In this article, I will not treat the developments among Western practitioners of Tibetan Buddhism.

10  There were no secular universities in traditional Tibet, only public and private schools.

11  See for instance Dreyfus (2003, pp. 10–13). See also Cabezón (1994, pp. 11, 13) and Kværne (2014, p. 85) for a discussion on scholasticism in Tibet.

12  There is also the religious tradition of the Bonpo, which even though distinct from Buddhism, has adopted many of its characteristics, especially when it comes to monasticism and education, as we will see later.

13  Namely, Gadong (dGa' gdong), Kyormolung (sKyor mo lung), Zulphu (Zul phu), Dewachen (bDe ba can), Sangphu (gSang phu) and Gungthang (Gung thang).

14  Personal information, WeChat discussion August 2019; this thesis is also reported by Dreyfus (2003, p. 6).

15  It is interesting to remark here that in the contemporary religious encampment of Larung Gar, in Serthar (Eastern Tibet), where nuns are trained alongside monks to become *khenmo*s, the debate part was only introduced into their curriculum in 2014 (and earlier for monks) and into the chart of their examinations in 2017, while the diploma and title itself were awarded since the early 1990s. See Padma'tsho/Baimacuo (2021, p. 16).

16  For more on Tibetan nuns' ordination and the different debates surrounding it see Mrozik (2009); Schneider (2012) and Price-Wallace and Wu in this volume.

17  Ms Rinchen Khandro is married to Mr Tenzin Choegyal or Ngari Rinpoche, the youngest brother of His Holiness the Dalai Lama; moreover, she served as Minister of Education in the Central Tibetan Administration in exile from 1993 to 2001.

18  See Karma Lekshe Karma Lekshe Tsomo (1988) for further information. Several international foundations with the aim of supporting Tibetan Buddhist nuns were set up at the same time or shortly after. To name just a few, these are: the Jamyang Foundation, especially reaching to nuns from the Indian Himalayas, founded by Venerable Karma Lekshe Tsomo, a Tibetan Buddhist nun from Hawai (and co-founder of Sakyadhītā); the Gaden Choling Foundation from Toronto supporting in particular nuns from Zangskar; and Tsoknyi Humanitarian Foundation taking care of nuns from Nangchen (Tibet).

19  The eclectic or non-sectarian approach in Tibetan Buddhism was promulgated by religious masters of the nineteenth-century *rimé* movement who wanted to come to an end with sectarian quarrels. In exile, the *rimé* approach is supported by many masters and above all by the Fourteenth Dalai Lama himself.

20  The Institute for Buddhist Dialectical Studies, also located in Dharamsala, was founded in 1973. For more information on its monastic training, see Lobsang Gyatso (1998) and Kværne (2014).

21  One idea behind secular education is also that some nuns might not stay nuns all their life and thus might adapt to lay life in future. Even though this has proved to be true, the subject is rarely talked about openly.

22  Within the framework of the "Mind and Life" exchange program, initiated by the Fourteenth Dalai Lama and Emory University in the USA, monks and nuns are invited to take part in annual workshops to study sciences for three years. Nuns from different nunneries have been participating in these workshops since 2011. At the end of this training, they receive a diploma.

23  Dolma Ling has a media centre where two nuns are working full-time and others temporarily.

24  On the role and value of memorisation in the context of monastic education, see Dreyfus (2003, pp. 91–97).

25  For studies on debate, see Perdue (1976); Dreyfus (2003); Liberman (2007) and Lempert (2012).

26　For more information on the traditional "winter debate," see Dreyfus (2003, pp. 234–36) and http://web.archive.org/web/2010 1203161011/http://www.qhtb.cn/buddhism/view.jsp?id=171 (accessed on 25 July 2022).

27　For more information on full ordination in Tibetan Buddhism, see Price-Wallace and Wu in this volume.

28　For more information on this aspect, see Schneider (2012).

29　Some more Western nuns have studied in the Institute for Buddhist Dialectical, but up to then, none of them had gone so far in their studies.

30　Personal communication from Kelsang Wangmo (Kerstin Brummenbaum). For more information on the German *geshema*, see her biography: Siegel (2017).

31　Since 1963, the heads and representatives of the four major schools of Tibetan Buddhism and the Bon tradition meet regularly in order to decide on major issues pertaining to religion.

32　She and her husband, Ngari Rinpoche, like Pema Chhinjor had been involved in the establishment of the Tibetan Youth Congress, the biggest Tibetan NGO in exile.

33　Personal communication from Rinchen Khandro (August 2012), which was confirmed by Thupten Tsering, joint secretary at the Department of religion and culture at this time.

34　In Tibetan, respectively: Byang chub chos gling, mKa' spyod dga' 'khyil gling, dGe ldan chos gling, and 'Jam dbyangs chos gling. For more information on nuns preparing the *geshema* examen in Khachö Gakyil Ling, see Plachta (2016) and Ehm (2020).

35　In Tibetan: Bod kyi btsun dgon dang slob gnyer khang gi gzhung chen bslab pa mthar son btsun ma rnams la dge bshes ma'i rgyugs sprod dang lag 'khyer 'bul phyogs kyi sgrigs gzhi/.

36　Up to then, nunneries did not follow exactly the same program as stipulated in the constitution. Thus, the new rules obliged them to adapt some of their courses.

37　Examinations are organised by turn in the different nunneries and each year during the fifth month (later changed to the eighth month); moreover, they are oral and written with three hours for principal subjects and three hours for secondary subjects, as well as a fifteen-minute test of debate.

38　Śākyaprabha (8th century) was one of the early translators and commentators on the Buddha's teachings. He was a disciple of Śāntarakṣita, the famous Indian master who ordained the first Tibetan monks, and a crucial link in the *Vinaya* tradition which is followed in Tibet. See Gardner (2019) and URL: http://www.rigpawiki.org/index.php?title=Shakyaprabha (accessed on 25 March 2022).

39　Personal communication from Geshe Rinchen Ngödrub (15 August 2012). Interestingly, Georges Dreyfus (2003) questions the relevance of *Vinaya* studies in the monastic curriculum altogether saying that "these texts contribute little to the intellectual qualities most valued by Tibetan scholars" and pointing to the fact that the "actual organisation of the order in Tibet derives not from the *Vinaya* but from the monastic constitution" (*bca' yig*), which gives the rules elaborated by each monastery itself. For more on monastic constitutions, see Jansen (2018).

40　Moreover, the *Gandentripa*, Thubten Nyima Lungtok Tenzin Norbu, was the very first to hold this position who is ethnically not a Tibetan, but a Ladakhi, even though he did his studies in Tibet and in Tibetan monasteries in exile.

41　These are the same nunneries as stated above: Jangchub Chöling located in Mundgod, Khachö Gakyil Ling situated in the Katmandu valley and Geden Chöling, Jamyang Chöling and Dolma Ling in the area of Dharamsala.

42　It might have been either a device used purposely by the director of TNP who comes originally from Kham herself or because of elderly *geshe*s not willing to participate in the event.

43　Originally, the graduation ceremony should have taken place in Dharamsala, at the main temple; however, it was decided finally that it would be more suitable to held it in south of India, where more Tibetans would be able to attend.

44　At the time of finalising this article (August 2022), ninety-three nuns from six nunneries (for the first time, a nun from Changsem Ling [Byang sems gling] in Kinnaur takes part) are passing their *geshema* examen in Geden Choeling nunnery, Dharamsala. Two of the examinators have come from Drepung monastery, two from Ganden monastery and the questions and corrections are dealt by another two *geshe*s from Sera monastery. The Geshema committee comprising two *geshema*s, a nun and a *geshe* teacher from Dolma Ling supervise the whole procedure.

45　Many photos and congratulations were sent through the social media application WeChat, largely used in China; however, it has been banned in India since June 2020 and can no longer be accessed.

46　To my knowledge, some *geshema*s are now teaching in Jamyang Choeling nunnery as well.

47　When visiting Tibet in 2018 and 2019, I saw nuns from Dragkar nunnery and Lamdrak nunnery debating in Kandze (Eastern Tibet); they were then studying the *pharchin* (Perfection of Wisdom) part of the curriculum. I learnt that in Ngaba, Gelukpa nuns are also studying in order to become *geshema*s.

48　In the past, Bonpo scholars used to join one or other of the Gelukpa monastic universities in order to deepen their knowledge, with some also passing the *geshe* degree. One of them was the eminent Professor Samten Karmay, who later became a researcher at the CNRS in Paris.

49　Personal communication from Kalsang Norbu Gurung (January 2020). *Geshe* Söpa Gyurme went back to Tibet in the early 2000s.

50    Personal communication from Kalsang Norbu Gurung (January 2020). According to Chech (1986, p. 11), the curriculum of the Bonpos in Menri monastery lasted eight years in 1986; it has been expanded over time to thirteen years (Ramble 2013, p. 7).

51    In the nunnery of Ratna Menling, the study program actually covers a total of eleven years, but because this was the first group of nuns to obtain the *geshema* degree, it was decided to expand the first two years (covering the topics of *düdra* [*bsdus grwa*] and *tshema* [*tshad ma*]) to four years. Personal communication from *geshema* Phuntsok Tzulzin (August 2022). See also Central Tibetan Administration (14 March 2022) and https://ybmcs.org/redna-menling-nunnery/ (accessed on 15 July 2022).

52    Edited by bLa rung ārya tāre'i dpe tshogs rtsom sgrig khang (2017).

53    I have been visiting and studying Tashi Gönsar since 1999, initially as part of my Ph.D. (Schneider 2013).

54    However, opinions are diverging on this matter, some also thinking that the ordination status has nothing to do with academic degrees. Personal communication from Khenpo Tenkyong (August 2022). For instance, in Tibet (Larung Gar and Tashi Gönsar), the lacking *gelongma* status has not been a problem.

55    Personal communication from Khenpo Tenkyong (March 2022).

56    See (Tibetan Nuns Project 2022) and (Voice of Tibet 2022); https://www.youtube.com/watch?v=J57KHRjx0l8 (accessed on 30 July 2022).

57    Personal communication from Khenpo Yeshe Tsering (August 2022).

58    Conference given at the Institute of Oriental Languages and Civilizations, 14 September 2016 (https://www.youtube.com/watch?v=LFUzfku_nzg (accessed on 30 March 2022)).

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
