# Peer review of "A Revolution in Red Robes: Tibetan Nuns Obtaining the Doctoral Degree in Buddhist Studies (Geshema)"

_religions, doi:10.3390/rel13090838_

Round 1

Reviewer 1 Report

This is a detailed and good study of "Tibetan Nuns Obtaining the Doctoral Degree in Buddhist Studies." It's a study from a cultural anthropological perspective, not an in-depth philosophical or religious one, but the facts are well researched, summarized, and estimated.

I doubt that the four Figures are necessary for the paper, but if it is common in the Religions, they should be left as they are.

If the entire titles of the author's papers were shown, it would make no sense to hide the author's name as xxx.

typos:

l.215: Prajñāparamitā>Prajñāpāramitā

l.502: geshemas>geshemas

Author Response

First of all, thank you very much for reviewing my article. I corrected the two errors and I will keep the photos having been trained as anthropologist who likes to show also visual aspects of Tibetan culture.

Reviewer 2 Report

This is a significant and timely article for the special issue onGender Asymmetry and Nuns’ Agency in the Asian Buddhist Traditions.” Now that recent articles have been published about the khenmo degree (female cleric-scholars) in Nyingma institutions, one in Religions, this genealogy into the emergence of the geshema degree among the Geluk is a much needed complement. The article pays close attention to the global context and major Tibetan actors, the development and implementation of the geshema degree, as well as the benefits to nuns, their enhanced status and vocational options following graduation, and their positive reception among lay Tibetans and monastics.

A few specific recommendations:

1.  In the introduction, the article should engage with Suzann Mrozik’s article, “A Robed Revolution: The Contemporary Buddhist Nun’s Movement” since it is alluded to in title. At minimum, the author could mention differences between two aspects of the nun’s movements with respect to Tibetan nuns: seeking full ordination vs. the advancement of nun’s education.

2. When first mentioning “religious encampments” (chos sgar) in eastern Tibet, the author could mention the Nyingma nature of these, and distinguish between temporary encampments of the past and more established institutions today, such as Larung Gar.

3. In the paragraph starting with “The path to equal education with monks was not without its pitfalls for nuns,” the author should be careful not to present equal education as a fait accompli. There have been significant advancements, for sure, but these are only at select institutions, not yet widespread. This point is made clear in the conclusion to the article.

4. Really interesting the discussion in “Seeking Recognition,” about the various conversations, actors, point of agreement and contention. How would the author characterize this: is it mostly the mobilization of high monastic allies? To what degree did women leaders in the Tibetan Nuns Project contribute to and set the agenda for discussions? What about the voices of nuns?

5. I also wonder if the author could speak to any differences between the Dalai Lama’s support but inaction around full ordination versus his clear endorsement and intervention to promote the advancement of nun’s education. Is it less controversial or more aligned with his own priorities?

6. When introducing the Tibetan Nuns Project, it might be helpful to give a broader context of international support for nuns. It’s good the author mentions Sakyadhita and its international nun’s movement, but there is also the appeal of nun-focused project with international patrons, such as the nuns in Nangchen (Tsoknyi Foundation), Lhadakh (Jamyang Foundation), etc.

These are all points of clarification which could be resolved by the addition of a few sentences on each point. The comparisons with the khenmo degree and the advancement of nun’s education outside of the Geluk school in the conclusion add significantly to the article. That may deserve a section of its own a bit sooner in the discussion of context. Overall excellent research on an important topic.

Author Response

First of all, thank you very much for reviewing my article and the many suggestions. I accepted them all and integrated them either in the text or in notes when I considered that it might affect the flow of reading.

Reviewer 3 Report

This article focuses on the Tibetan nuns obtaining the doctoral degree in Buddhist studies, which is an important topic. There are several interesting ethnographic observations in this article and some valid analytical points. However, for the points of "Revolution in Red robes" and  " this means for the future of the position of women in the religious sphere", the author need to do more logical analysis based on the literature review and field evidences. 

Author Response

First of all, thank you very much for reviewing my article.

I have thought about revising the title, but was not really convinced about the necessity. I have also thought about either revising or leaving out the sentence “this means for the future of the position of women in the religious sphere”, but was not really convinced. I could of course nuance it and put “maybe” in the middle, but I think in the conclusion I give already some future tendencies who point to the fact.

I hope you do not mind.